# Degradation of Poly(ε-caprolactone) by a Thermophilic Community and *Brevibacillus thermoruber* Strain 7 Isolated from Bulgarian Hot Spring

**DOI:** 10.3390/biom11101488

**Published:** 2021-10-09

**Authors:** Nikolina Atanasova, Tsvetelina Paunova-Krasteva, Stoyanka Stoitsova, Nadja Radchenkova, Ivanka Boyadzhieva, Kaloyan Petrov, Margarita Kambourova

**Affiliations:** 1Institute of Microbiology, Bulgarian Academy of Sciences, Acad. G. Bonchev Str. Bl. 26, 1113 Sofia, Bulgaria; nikolina@microbio.bas.bg (N.A.); pauny@abv.bg (T.P.-K.); stoitsova_microbiobas@yahoo.com (S.S.); nstoicheva@yahoo.com (N.R.); petrovaim@abv.bg (I.B.); 2Institute of Chemical Engineering, Bulgarian Academy of Sciences, Acad. G. Bonchev Str. Bl. 103, 1113 Sofia, Bulgaria; kaloian04@yahoo.com

**Keywords:** plastic biodegradation, bacterial decomposition of poly-ε-caprolactone, thermophilic PCL degrading community, *Brevibacillus thermoruber*, microbial biofilm

## Abstract

The continual plastic accumulation in the environment and the hazardous consequences determine the interest in thermophiles as possible effective plastic degraders, due to their unique metabolic mechanisms and change of plastic properties at elevated temperatures. PCL is one of major biodegradable plastics with promising application to replace existing non-biodegradable polymers. Metagenomic analysis of the phylogenetic diversity in plastic contaminated area of Marikostinovo hot spring, Bulgaria revealed a higher number taxonomic groups (11) in the sample enriched without plastic (Marikostinovo community, control sample, MKC-C) than in that enriched in the presence of poly-ε-caprolactone (PCL) (MKC-P), (7). A strong domination of the phylum *Proteobacteria* was observed for MKC-C, while the dominant phyla in MKC-P were *Deinococcus-Thermus* and *Firmicutes*. Among the strains isolated from MKC-P, the highest esterase activity was registered for *Brevibacillus thermoruber* strain 7 at 55 °C. Its co-cultivation with another isolate resulted in ~10% increase in enzyme activity. During a 28-day biodegradation process, a decrease in PCL molecular weight and weight loss were established resulting in 100% degradation by MKC-P and 63.6% by strain 7. PCL degradation intermediate profiles for MKC-P and pure strain were similar. Broken plastic pieces from PCL surface and formation of a biofilm by MKC-P were observed by SEM, while the pure strain caused significant deformation of PCL probes without biofilm formation.

## 1. Introduction

Plastic’s invasive entrance in human life notably improved its standard, making it easier, safer, and more colorful. Due to their exclusive properties, such as bio-inertia, resistance to environmental influence and microbial action, easy manufacturing, light weight, and low production cost, everyday plastic use shows an exponentially increasing trend for production and consumption. Synthetic plastics are widely used in the global economy, with an annual worldwide production at a scale of 350 to 400 million metric tons [1]. However, due to their nature of solid and densely cross-linked polymers, they are not easily accessible for microbes and enzymes. In the category of biodegradable polyesters, poly-e-caprolactone (PCL) is among the highly attractive compounds. For the forecast period 2019–2027, an annual growth of 11.2% is envisaged, reaching US$ 214.2 million by the end of 2021 and 402.7 million by an end of 2027 [2]. Europe and North America are expected to remain dominant regions throughout the period 2020–2030 with a market share close to 60%. Among other economies around the world, the COVID-19 pandemic has impacted the functioning of the global polycaprolactone market space [3].

Poly-Ɛ-caprolactone (C_6_H_10_O_2_)n is derived from the chemical synthesis of a crude oil prepared by ring-opening polymerization of ε-caprolactone and 2-methylene-1-3-dioxepane. It is semi-crystalline polyester whose crystallinity decreases with an increase in its molecular weight. PCL is characterized by a low melting point of around 60 °C and a glass transition temperature of about −60 °C. It possesses exceptional properties such as good hydrophobic characteristics, chemical, UV-, and wear resistance, low temperature flexibility, gloss, and adhesion. PCL-based products are an effective solution for different healthcare applications due to their non-toxic and non-hazardous nature, good biocompatibility and biodegradability [4]. They have numerous applications as elastic biomaterials, such as implants, orthopedic, surgical drapes, general-purpose tubing, drug delivery, wound dressing, suture, injection-molded devices, etc. The possibility for an extrusion of PCL filaments with an accurate diameter is of growing importance in 3-D printing technology due to its stable melting point and no bubble formation. PCL dentistry application includes dental splints and root canal filling. It is a preferable plastic in making automotive components, construction materials, packaging, jewelry, models, prototyping, hot-melt glue and laminating pouches. PCL-based polyurethanes are widely used in coatings, adhesives, sealants, and elastomers. Currently, PCL is still not a cost-effective polymer with a USD 6–8/kg market price. Great efforts have been focused on mass production at lower costs.

Biodegradation is a process of decomposition of a large polymer molecule by microorganisms to oligomers that could be utilized as a sole source of carbon and energy. Unlike the high biodegradability of the natural plastics, the synthetic polymers have a low biodegradability rate [5,6,7]. Although PCL refers to the group of biodegradable plastics, the reports on its biodegradation have shown that it is still a slow process. Several fungi, such as *Penicillium funiculosum*, *Aspergillus flavus*, *Rhizopus delemar*, *R. arrizus*, and *Candida cylindracea* [8] and bacteria such as *Tenacibaculum*, *Alcanivorax* and *Pseudomonas* [9], *Alcaligenes faecalis* [10], *Bacillus pumilus* [11], and *Clostridium acetobutylicum* [12] have been reported as PCL degraders.

According to Pinto et al. [13] the nature of the degrading organisms is among the most important factors for an effective biodegradation, together with the abiotic types of plastic and the physicochemical conditions. Biodegradation is assumed to occur in the amorphous phase of plastic polymers [14,15]. The molecules in the amorphous region are not able to move significantly at lower temperatures, suggesting a higher efficiency of thermophilic processes in which the polymer chains can gain enough mobility to access the enzyme active sites [16,17]. The changes seen in the plastic’s properties at an elevated temperature suggest improved bioavailability and solubility [14]. Thermophilic processes are characterized by a higher enzyme turnover rate as a result of the decreased polymer strength at enhanced temperatures, high diffusion rates of organic compounds, decreased viscosity of culture liquids, and reduced risk of microbial contamination. Several thermophiles have been reported as plastic degraders. They mainly belong to a bacilli group, such as the polyethylene degraders *Bacillus* sp. BCBT21 and *Brevibaccillus borstelensis* strain 707 [18,19], nylon 6 and 12 degraders *Anoxybacillus rupiensis* and *Geobacillus thermocatenulatus* [20,21]. Information concerning thermophilic PCL degradation is very limited and concerns only representatives of the genus *Streptomyces* [22,23]. Still, thermophilic bacilli able to degrade PCL are not known [24].

In the present paper, we report an effective PCL degradation at 55 °C by a microbial community from the Marikostinovo hot spring (MKC), Bulgaria and an isolation of *Brevibacillus thermoruber* strain 7 with an esterase activity.

## 2. Materials and Methods

### 2.1. Sample Collection and Culture Enrichment

Pooled samples containing plastic debris and water in the vicinity of plastic wastes were collected from five Bulgarian hot springs located in south-west Bulgaria: Rupi (41.45857° N, 23.26209° E), Levunovo (41.4872° N, 23.3055° E), Marikostinovo (41.4314° N, 23.3335° E), Simitli (41.8897° N, 23.1060° E), and Dolno Osenovo (41.9544° N, 23.2413° E). The temperature of the sites varied from 55 to 72 °C, with a pH in the range 7.1 to 8.8. Marikostinovo water characteristics were 57 °C, pH 7.3, low dissolved-mineral content (1031 mg/L) with a mineral composition (mg/L): F^−^, 8.4; Cl^−^, 18; SO_4_^2−^, 227; HCO_3_^−^, 412; SiO_3_^2−^, 77; Na^+^, 250; Ca^2+^, 23; Mg^2+^, 6. The samples were transferred to the laboratory in sterile glass bottles kept in a thermostat bag and 2 g plastic waste and 2 mL water used as inocula for enrichment in 20 mL basal medium containing: NH_4_NO_3_, 0.01%; KH_2_PO_4_, 0.03%; K_2_HPO_4_ 0.14%; MgSO_4_, 0.01%; FeSO_4_·7H_2_O, 0.002%; Na_2_MoO_4_·2H_2_O, 0.0005; yeast extract, 0.1%. As an only carbon source poly(ε-caprolactone) pearls, average Mw 80,000 (Sigma-Aldrich, Steincheim am Albuch, Germany) and a diameter ranged from 2.9 to 4.8 mm were used. PCL pearls were sterilized separately by three hours soaking in 96% ethanol and five minutes sonication at room temperature. They were added to the basal medium at a final concentration of 0.3 ± 0.02%. The cultivation was performed at 55 °C, 80 rpm for 72 h. The control flasks did not contain PCL. Growth (OD660) and esterase activity in the supernatant were monitored daily.

### 2.2. Microbial Community Analysis

Microbial composition of Marikostinovo community after cultivation in a basal medium without plastic (MKC-C) and in the presence of 0.3% PCL (MKC-P) was determined after 5 days-cultivation. After centrifugation of 200 mL culture liquid the pellet was sent to Eurofins Genomics Europe, Ebersberg, Germany for a metagenomic analysis. DNA was extracted with a commercial kit according to the manufacturer’s manual. 16S targets were PCR amplified from DNA extracts using target specific NGS primers and analyzed by Amplicon sequencing on the Illumina MiSeq platform. During the bioinformatics analysis sequences were sorted into sequence sets according to their similarity. Each set was represented by a master sequence. Comparison of each master sequence with entries in the nucleotide collection of the US National Center for Biotechnology Information, NCBI provided the taxons present in the samples. Taxons with a fraction of at least 0.1% of all assigned reads were reported.

### 2.3. Isolation of Pure Strains and Screening of Isolates

An MKC-P sample was spread on rich medium agar plates containing 0.2% peptone; 0.1% yeast extract; 0.1% glucose; 2% agar. The mixed cultures streaked at least three times for an isolation of pure strains and single colonies with a different appearance were selected for further work. The screening procedure for PCL degrading bacteria was performed as previously described [25] with some modifications. An emulsion of 1% PCL with average Mw 14,000 (Sigma-Aldrich, Steincheim am Albuch, Germany) in acetone was prepared on magnetic stirrer at 50 °C. Twenty mL of the emulsion were added through a sterile filter to the sterile basal medium supplemented with 2% agar. The magnetic stirring at 50 °C continued until acetone evaporated and the medium was poured into plates. One loop of each strain culture was placed on the agar surface and incubation continued for 24 h at 55 °C. Biodegradation activity was determined by the formation of clear halos around the growth spots.

### 2.4. Phylogenetic Characterization of the Isolated Bacteria

Petri dishes containing the screened pure strains were sent for phylogenetic characterization in Macrogen Europe BV, Amsterdam, Netherland where the full length 16S rRNA gene sequences were retrieved by using universal bacterial primers 27F and 1492R. Sequence length for all isolates was ˃1440 bp. They were aligned to organisms present in the Gene Bank database on 15 April 2021 using the Basic Local Alignment Search Tool (BLAST, https://blast.ncbi.nlm.nih.gov/Blast.cgi?PROGRAM=blastn&PAGE_TYPE=BlastSearch&LINK_LOC=blasthome, accessed on 29 January 2021 and 15 April 2021). Their DNA sequences accession numbers were MW541894 to MW541896 and MW927322 to MW927332.

### 2.5. Optimization of the Parameters for PCL Degradation

The effect of pH on the ability of the MKC-P, *B. thermoruber* strain 7, and co-culture of strain 7 and strain 2 to degrade PCL as a sole carbon source was investigated by the commonly used in PCL degradation esterase assay in pH range 6–9 (0.5 interval) at 55 °C. Cultures were incubated in the basal medium for 72 h on a shaker (80 rpm) at 55 °C. The optimum pH value was employed in the experiment for evaluation of the temperature effect on PCL degradation. The temperature dependence of PCL was followed at 50, 55, 60, and 65 °C. A control of an uninoculated culture medium with PCL was run at all temperatures. The influence of PCL concentration in the medium on the esterase activity was followed at 0.1, 0.3, 0.5, and 0.7% (*w*/*v*).

### 2.6. Esterase Assay

As PCL represents a polyester, esterase assay was used for an estimation of enzyme activity. It was measured in the supernatant after centrifugation of the culture liquid at 4000× *g* for 15 min. Hydrolysis of p-nitrophenyl palmitate (p-NPP) as a substrate was determined spectrophotometrically as described previously [26] at 55 °C in 0.05 M sodium phosphate buffer, pH 7.8 at 405 nm. One unit of esterase activity was determined as the amount of enzyme needed to liberate 1 μM p-nitrophenol per minute in the described conditions. A molar extinction coefficient for p-nitrophenol at pH 7.8 was found to be 3.39 × 10^3^/M.

### 2.7. Estimation of the Bacterial Biomass

It was impossible to use standard methods for the estimation of a bacterial growth-like measurement of optical density or microscopic counting due to the formation of visible biofilms on the surface of the PCL pearls during the 4-week biodegradation process in MKC-P. This is why the microbial growth was evaluated by measuring the protein concentration in the culture liquid and in the cells removed from PCL surface after SDS treatment. Colonized PCL pearls taken from the bacterial liquid culture were washed with water and then treated with 2% SDS at 40 °C for 4 h. The procedure was repeated, and the cells were harvested by centrifugation. Furthermore, they were washed in 0.06 M potassium phosphate buffer (pH 7.8) and suspended in five-times their wet volume of the same buffer. The suspension cells were disrupted in an IsoLab sonicator at 40 kHz frequency for two periods of 5 min at room temperature. Cell debris were removed by centrifugation at 12,000× *g* for 10 min and protein content in the resultant supernatant determined by the method of Lowry et al. [27], with bovine serum albumin as a standard. The esterase activity in both supernatant and cell debris suspension was also determined.

### 2.8. Estimation of the Gravimetric Weight

PCL biodegradation was determined by the gravimetric method for 4 weeks using a balance with 10 μg accuracy. The initial weight of PCL pearls in all flasks was 150 ± 9 mg, and the weight loss was calculated each week for three of the flasks. The culture medium was changed weekly in the flasks, in which the weight was not determined. To assure accurate measurement of the residual weight of the pearls, the bacterial biofilm was removed from PCL surface by 2% (*v*/*v*) SDS as described above. The pearls were washed with water and ethanol and dried on a filter paper overnight at 50 °C. The weight loss of PCL pearl weight per day (degradation rate) was calculated on the base of their gravimetric weight loss according to the formula: DR = (W_0_ − W)/7 [18], where DR is the degradation rate, W_0_ is the PCL pearl weight at the end of the previous week (mg), and W is the retention pearl weight at the end of the current week. Three flasks were used for each week measurement and average weight change was estimated. The standard error (SE) calculated from data deviation was 8%.

### 2.9. Gel Permeation Chromatography

GPC analysis was performed in the different stages of the degradation process with Mw 14,000 PCL. Supernatant (3 mL) was frozen in 30 mL glass vial at −15 °C and kept in lyophilic dryer at −60 °C with 80 mTor vacuum for 12 h. The sample was dissolved in 1 mL HPLC grade tetrahydrofuran (THF) for 2 h at 45 °C. The sample was filtered through 0.45 μm PTFE syringe filter in standard 2 mL chromatography vials. A GPC analyzer (Model SHIMADZU Nexera) (Shimadzu, Kyoto, Japan) equipped with 5-Channel Degasser DGU-20A, HPLC Pump LC-20AD, Autosampler SIL-20AC, Column Oven CTO-20AC, and Refractive Index Detector RID-20A was operated at 45 °C and a THF elution rate of 1.0 mL/min. The GPC Column Set PSS SDV 50 Å (300 mm × 8.00 mm × 5 μm), PSS SDV 100 Å (300 mm × 8.00 mm × 5 μm), PSS SDV Linear M (300 mm × 8.00 mm × 5 μm) were used. Shimadzu, Kyoto, Japan)

### 2.10. Scanning Electron Microscopy (SEM)

MKC-P and the selected strain were inoculated into 20 mL of basal medium with 0.3% PCL as a sole carbon source. Cultivation was performed for 72 h or 3 weeks. Control samples were processed in parallel, in the absence of bacteria. The samples were fixed for 2 h in 4% glutaraldehyde in 0.1M Na cacodylate buffer (pH 7.2), then washed and post-fixated in 1% OsO4 for 1 h. Dehydration was performed in graded ethanol series. After sputter-coated with gold using Edwards’s sputter coater, the samples were examined by SEM (Philips scanning electron microscope, Phillips, Amsterdam, The Netherlands), at accelerating voltage 30 kV.

## 3. Results

### 3.1. Phylogenetic Diversity in Samples Cultivated with and without Plastic

An esterase activity of pooled samples collected from five Bulgarian hot springs and enriched in a basal medium with PCL was registered in two of the samples, from Levunovo (54.83 U/mL) and from Marikostinovo (174.9 U/mL). The highest activity for both samples was reached after 48 h of cultivation and did not change during the week of the experiment. Marikostinovo community (MKC) was selected for further work. Metagenomic analysis revealed that the community composition in the control that did not contain plastic (MKC-C) (Table 1 and Figure 1A) was characterized by a higher diversity (11 phylogenetic groups, 5 orders) and a strong domination of the phylum *Proteobacteria* (81.2%), mainly *Betaproteobacteria* (77.2%). Other represented phyla were *Deinococcus-Thermus* (15.5%) and *Firmicutes* (2.3%). Representatives of *Archaea* were not identified. Community composition in the presence of PCL (MKC-P) (Table 2 and Figure 1B) was characterized by a limitation of the community diversity (7 phylogenetic groups, 3 orders) and a strong domination of the phyla *Deinococcus-Thermus* (65.6%) and *Firmicutes*, Class *Bacilli* (30.7%). The increase in the order *Bacillales* share was more than thirteen-fold and of *Deinococcus-Thermus* more than four-fold. Appearance of noticeable fractions of *Meiothermus* (58.1%) and *Brevibacillus* (18.5%) could suggest their active participation in PCL degradation. Sharp decrease of *Proteobacteria* reads represented by one species, *Elioraea tepidiphila* (3.7%) was observed. Two community metagenomes were deposited in NCBI by BioProject accession number PRJNA766622 (https://www.ncbi.nlm.nih.gov/sra/PRJNA766622, accessed on 29 September 2021).

### 3.2. Isolation of PCL Degrading Bacteria

Fourteen visibly different colonies isolated from MKC-P were sequenced and identified. They referred to four thermophilic species from three families, nine strains from *Brevibacillus thermoruber* (*Paenibacillaceae*), two strains from *Aneurinibacillus thermoaerophilus* (*Paenibacillaceae*), two strains from *Geobacillus thermodenitrificans* (*Bacillaceae*), and one strain from *Meiothermus cateniformans* (*Thermaceae*).

Pure strains were checked for a formation of clear zones around the colonies on agar plates in a basal medium with PCL as a sole carbon source (Figure 2). A good growth and halos were observed for *B. thermoruber* strain 7, *A. thermoaerophilus* strain 2 and *M. cateniformans* strain 12. This result is in good agreement with the appearance of noticeable fractions of *Brevibacillus* and *Meiothermus* registered by the metagenomic analysis.

The esterase assay confirmed quantitatively an activity for the three pure strains. The highest activity was measured for strain 7. Cells of strain 7 were deposited in NBIMCC (WDCM No. 135) under the number 9076. Some moderate activity was observed for strain 2 and strain 12 (Table 3). Co-cultivation of strain 2 and strain 7 resulted in higher activity, while the esterase activity of co-cultures of strain 7 and strain 12 was even lower than that of strain 7 alone, determining the elimination of strain 12 from further experiments.

### 3.3. Optimization of Physico-Chemical Parameters for PCL Degradation

Optimization of the degradation conditions revealed highest esterase activity of MKC-P and *B. thermoruber* strain 7 at pH 7.5–8.0, while it was 8.0–8.5 for the co-culture of strain 7 and strain 2. This result is in a good coincidence with the slightly alkaline pH of the spring water. The enzyme activity was enhanced with temperature reaching a maximum at 55 °C for all variants. Temperature increases above 55 °C resulted at plastic melting especially well, demonstrated at 65 °C where an activity was not registered. Comparison among pNP palmitate (C16) and pNP caprylate (C8) as a substrate in the esterase assay revealed that activity with pNPP was 1.34-fold higher in the culture liquid from MKC-P and 1.2-fold higher in the case of strain 7.

The influence of the substrate concentration in the medium is shown on Figure 3. Highest enzyme levels were measured at PCL concentration of 0.3%, 375 U/mL for MKC-P, 290 U/mL for strain 7, and 325 U/mL for co-culture of both pure strains.

### 3.4. Characteristics of PCL Biodegradation Process

The extent of biodegradation of 0.3% PCL (Mw 80,000) by MKC-P, strain 7, and co-culture of strain 7 and strain 2 was followed for 4 weeks in liquid culture at 55 °C, 80 rpm (Table 4).

Full degradation of PCL and lack of flake traces was observed after four weeks in the presence of MKC-P, while the gravimetric weight loss was 63.6% in the case of strain 7 alone and 74.9% by the co-culture of strain 7 and strain 2. The constant PCL weight in the control flasks demonstrated a lack of self-degradation throughout the experiment. A fast degradation rate with a maximum of 8.83 mg/d was observed at the beginning of the process by MKC-P, while the highest levels of ~4 mg/d were measured at the second part of the process when pure strains were used. At the end of the first week the highest activity of 375 U/mL was measured in the culture liquid for MKC-P, the lowest activity 290 U/mL for strain 7 and an average activity of 325 U/mL for the co-culture and showed a slight tendency for a decrease during that time. The enzyme was extracellular as it was not found in cell debris. The difference in enzyme activity among the variants was not as significant as the degradation rate was and the higher efficiency of the degradation process by MKC-P was explained with the biofilm formation that posed microorganisms and substrate in a close vicinity. Cell concentration determined as a protein in the biofilm was the highest in MKC-P, reaching 7.3-fold higher levels in comparison with the other two samples. The low protein concentration in the adherent cells from strain 7 and co-culture with strain 2 suggested that these two strains were not main participants in the biofilm formation.

The degradation rates in a process with lower PCL concentration of 0.1% were followed. Measured values were 9.96 mg/d for the first week, 6.65 mg/d for the second week, 4.42 for the third week and the whole process finished a week earlier for MKC-P. Consequently, a three-fold decrease in PCL concentration up to 0.1% resulted in only ~10% increase in the maximal degradation rate. Similar were the values for strain 7 (2.67, 3.15, 5.2, and 4.4, respectively). The degradation rate in experiments with PCL Mw 14,000) was 10.24 mg/d for first week; 7.5 mg/d for the second week; 3.69 for third week and at the end of the fourth week no PCL flakes were observed. These values were similar to the values when PCL Mw 80,000 was used.

Penetration chromatography analysis of PCL intermediates revealed a different elution pattern for MKC-P, strain 7 and co-culture of strain 7 and strain 2 that was especially well demonstrated on 48 h of cultivation (Figure 4). A peak corresponding to the molecular weight of 14,000 representing the original PCL (Figure 4A, Control) added to the culture medium appeared in the elution volume of 18.45 mL in strain 7 and co-culture of strain 7 and strain 2 samples, however with a different share—83.74% for strain 7 and 54.99% for the co-culture. In the supernatant from strain 7, the only additional peak was with an elution volume of 27.06 mL that corresponded with the monomer Ɛ-caprolactone (14.99%). In the co-culture of strains 7 and 2, three other peaks with elution volumes of 24.39, 25.13, and 26.93 mL were observed, corresponding of intermediates with a molecular weight of hexamer (22.78%), trimer (3.67%), and monomer (4.25%) (Figure 4B). In the supernatant from MKC-P, PCL peak was almost not observed and the registered peaks with elution volumes 23.82 mL and 25.79 mL corresponded with 12-mer (8.2%) and dimer (91.7%). The characteristic peaks in the elution profiles of the three samples were very similar for the samples from 72, 96, 120, and 168 h of cultivation.

### 3.5. SEM Investigations

The changes in the plastic surface were confirmed by SEM. Control samples showed that the surface of the PCL pearls was not completely smooth (Figure 5A) and contained some wrinkles and deformations. No apparent effect of incubation duration was noted. Cultivation of *B. thermoruber* strain 7 for 72 h resulted in the occurrence of small bubble-like elevations on the surface of PCL pearls (Figure 5B) and shallow deformations (Figure 5C,D), which did not differ considerably from deformations present on the control pearls (Figure 5A). Adherent bacterial cells measuring c.a. 1.1 × 0.6 µm^2^ were only occasionally present (Figure 5D). The 3-week cultivation in the presence of strain 7 resulted in significant changes of PCL appearance which expand throughout the pearls surface. Characteristic was the formation of shallower (Figure 5E,F) or deeper (Figure 5G,H) infolds. The presence of adherent bacteria was still only occasional (Figure 5H), which implies that the dramatic changes on the PCL surface were much more likely due to the release of active bacterial products in the culture medium and not to biofilm formation.

The SEM picture of the PCL cultivated in the presence of the bacterial community differed substantially from the one for the single strain. At the earlier tested interval of 72 h, foci of adherence of bacterial cells (Figure 6A,B) were observed, and the development of biofilm microcolonies (Figure 6C,D) was frequent. At least three different morphotypes of bacteria were present in the microcolonies. They were distinct in width, length and the width to length ratio. The first type had a size of (1.05 ± 0.13) × (0.63 ± 0.10), length/width ratio 1.6 (Figure 6G) and were morphologically similar to those of strain 7 (Figure 5D). They represented 25% of all measured cells. The second type was the most abundant (68%), with a size of (1.45 ± 0.29) × (0.45 ± 0.05), length/width ratio 3.22 (Figure 6G–I). Only a few bacteria with the third type of registered morphology were visible in the biofilm. They were pear-shaped cells, with a size of (1.07 ± 0.12) × (0.45 ± 0.02), length/width ratio 2.4 and represented 7% of the measured cells (Figure 6I). On week 3, the incidence of significantly deformated loci did not considerably increase in comparison with the 72-h interval. Part of the bacteria tended to be submerged under the surface of the plastic pearl. Grainy structures appeared and bacterial cells were submerged under the grain-like debris (Figure 6J,K). This grainy material could be composed of the extrapolymeric substance of the biofilm, with a possible contribution of undecomposed plastic fragments.

## 4. Discussion

During initial enrichment of MKC in the basal medium, a low growth (OD660) was observed for both MKC-P and MKC-C, however a formation of a colored biofilm on the plastic surface in PCL presence as a carbon source was visible. Metagenomic analysis of the two samples revealed that the biodiversity in MKC-P was lower (7 taxonomic groups) than that of MKC-C (11 taxonomic groups). Thermophilic *Proteobacteria* representatives dominated in MKC-C. *Proteobacteria* presence is universal for the environmental samples [28]. Its significant reduction from 81.2%, 7 phylogenetic groups in MKC-C to 3.7%, only one species in MKC-P resulted in flourishing of *Meiothermus* and *Brevibacillus* in MKC-P that suggested an active participation of these genera in the degradation process. Similar domination of polymer-active taxa in plastic containing samples has been reported for some marine communities [29,30]. Unfortunately, a comparison among the composition of thermophilic communities with and without plastic is not possible due to the lack of such information.

Fourteen pure strains were isolated on rich medium petri dishes after enrichment in the basal medium with PCL. Among them three strains demonstrated esterase activity, two belonged to an order *Bacillales* and one to an order *Thermales*. The isolation of two PCL degrading bacilli supported the opinion that thermophilic members of the family *Bacillaceae* are a good source of bacteria for bioprocessing and biotransformation [20]. The strain with the highest activity was identified as belonging to the species *Brevibacillus thermoruber* and was designated as a primary degrading microorganism and represented the first member of bacilli group able to degrade PCL. The genus *Brevibacillus* is known as an object of great biotechnological interest [31], however up to now only its activity toward low density polyethylene (LDPE) has been reported. The thermophilic bacterium *B. borstelensis* has been able to degrade 11% of polyethylene for 30 days [19]. *B. borstelensis* has not been good in the biofilm formation unlike another LDPE degrader *B. parabrevis* [32]. During the first week of cultivation of the strain 7 with PCL, low degradation rate of the plastic was registered, and together with this SEM analysis did not show considerable adherence of bacteria on the PCL pearls. As the rate of plastic degradation as a rule is low, it is not necessarily a presence of the primary degrader in high cell numbers. This could explain the lower concentration of *Bacillaceae* (among which is strain 7) in comparison with *Meiothermus* established in MKC-P by the metagenomic analysis. Such a low concentration of the primary degrader *Streptomyces thermonitrificans* PDS-1 was observed in the simultaneous cultivation with *Bacillus licheniformis* HA1 as a degrader of the intermediates [22]. The domination of *Meiothermus* could suggest its participation mainly in biofilm formation and/or in the utilization of the degradation intermediates. It is possible that not all microorganisms in the community involved in plastic degradation were detected and/or cultivated.

The effective plastic degradation by thermophilic bacilli have been reported for *Anoxybacillus rupiensis* Ir3 growing very well on nylon 6 [20] and *Geobacillus thermocatenulatus* that has decreased the molecular weight of nylon 12 and 66 for 20 days [21]. The degradation process with *Clostridium thermocellum* resulted in 60% weight loss of 50 mg amorphous PET film for 14-days reaching a degradation rate of 2.2 mg/d [33]. *Thermobifida fusca* has decreased an average Mw of PET with 50% for 3 weeks [34].

Several mesophilic bacteria have been reported as PCL degraders. Weight loss and surface erosion of PCL degraded by *Bacillus subtilis* has been observed after 14 days of cultivation [15]. Biodegradation of PCL by a mesophilic *Bacillus* sp. has been seen within 20 days of cultivation at 40 °C although some surface changes have been visible after 7 days [11]. The representatives of the genera *Pseudomonas*, *Alcanivorax*, and *Tenacibaculum* isolated from deep seawaters at Toyama and Kume, Japan have caused morphological changes of PCL fiber surface after week cultivation at 25 °C [9]. The residual breaking strength of PCL fibers reached 0% after 9 months soaking in deep seawaters. *Pseudomonas* and *Rhodococcus* sp. along with two fungal strains have degraded PCL films up to 53% (*w*/*w*) in 30 days of incubation [35].

Previously reported thermophilic bacteria active on PCL have been related to the genus *Streptomyces*. *Streptomyces* degraders have been active at 40–50 °C and have had a terrestrial origin. *Streptomyces thermonitrificans* PDS-1 has been isolated from compost of the fish-processing industry [22], and *Streptomyces thermoviolaceus* subsp. *thermoviolaceus* 76T-2 has been isolated from soil in Taiwan [23]. Strain 7 was isolated from a hot spring and showed the highest temperature for degradation (55 °C), which is close to the melting temperature of PCL (60 °C) at which amorphous regions become soft and flexible. Unlike most of the reported mesophilic and thermophilic degraders, in our experiments visible changes in PCL surface were observed at 72 h of MKC-P and strain 7 cultivation. A comparatively fast degradation process with MKC-P and strain 7 and early visible changes in plastic surface confirmed that the isolate was among the most effective mesophilic and thermophilic PCL degraders isolated so far. The effectiveness of the biodegradation process was highest when MKC-P was used in comparison with strain 7 alone and co-cultures of strain 7 and strain 2. The available information for the higher efficiency of plastic degradation by the microbial community in comparison with single isolates or a combination of two strains is scarce [36]. A microbial consortium isolated from plastic garbage processing areas has been significantly more effective in a gravimetric weight reduction and Mw reduction of LDPE than three pure strains isolated from the same consortium [37]. The gravimetric weight loss of PCL was 41.2% by MKC-P and 10.2% by strain 7 after seven-day process and 100% and 63.6% correspondingly after 4-weeks. The highest degradation rate was 4.42 mg/d for strain 7 and 8.83 mg/d for MKC-P. MKC-P fully degraded 0.3% PCL for 4 weeks and 0.1% PCL for 3 weeks. An enhanced biodegradation potential by a combination of two bacterial species has been reported for *Streptomyces thermonitrificans* PDS-1 and *Bacillus licheniformis* HA1 [22]. *S. thermonitrificans* PDS-1 has been established to be responsible for primary PCL degradation in a fermentor seeded with raw compost materials, while *B. licheniformis* HA1 has had no primary degradation capability for PCL and its oligomers. The degree of PCL decomposition has been only 5% by the action of PDS-1 alone, and 16% of the mixed culture containing both thermophiles. A synergistic effect has been suggested as a reason for the accelerated PCL degradation that has involved the consumption of ε-caprolactone by the coexistent microorganism promoting the growth of the primary degrader. The presently investigated combined use of strain 7 and strain 2 resulted in only a 10% increase in the degraded polymer, probably as a result of the simultaneous action of enzymes by both organisms, but not as a synergistic effect of degradation. The effectiveness of the degradation process by MKC-P and strain 7 at different PCL concentration did not differ significantly in the presence of 0.1% compared with that at 0.3% PCL. Although a strong dependence of the plastic degradation process from its molecular weight is commonly accepted [4] probably the insignificant difference in the case of PCL (Mw 14,000 and 80,000) was connected with the fact that an increase in its molecular weight resulted in an increase of the amorphous regions [16].

Gel penetration chromatography analysis gave evidence that MKC-P is the most effective in PCL degradation as PCL did not accumulate in the supernatant but was assimilated by the community members. The monomer (ε-caprolactone) was also not detected, suggesting that it was consumed by an organism in the community. Registration of the monomer in the supernatant from strain 7 suggested that this microorganism was unable to assimilate ε-caprolactone or its assimilation was slower than its liberation from the PCL molecule. The registered monomer suggested that ester bonds are scissored at the ends of the polymer chain and the enzyme from this strain is an exoenzyme. Appearance of variety of low molecular weight intermediates in the supernatant from the co-culture of strain 7 and strain 2 confirmed the suggestion that strain 2 synthesized enzyme(s) different from that of strain 7 that act close to the end of the polymer chain. Unlike our results *Streptomyces thermonitrificans* PDS-1 has been reported as the only primary degrader, however its co-cultivation with *Bacillus licheniformis* HA1 has resulted in an accelerated PCL degradation and an increased deal of the decomposed polymer as HA1 has degraded the intermediates [22]. Esterases and particularly lipases were reported as main enzymes involved in the degradation of the polyester PCL [4,38,39]. Higher enzyme activity of MKC-P and strain 7 registered with pNP palmitate as a substrate than that with pNP caprylate suggested that the strain synthesized a lipase. Further detailed characterization of all enzymes included in the degradation process and the mechanism of their action is an object of a further work.

The highest degradation rate in flasks inoculated with the community in the early stage of the process could be explained by the presence of microorganism(s) that could contribute attachment to the plastic surface and biofilm formation. SEM results implied that this biofilm contained at least three different strains of bacteria. One of them was morphologically similar to the few cells, adherent to PCL in the sample treated with *B. thermoruber* strain 7 alone. As a single strain, strain 7 produced no biofilm, and could be characterized as adhesion deficient. However, the presence of the other stains in the community apparently increased its biofilm proficiency. Notably, cells of the “strain 7 morphotype” accounted for 25% of all bacteria in the biofilm formed by the community on hour 72. This indicates that there could be synergistic interaction between the community strains in the colonization of the plastic surface. Synergistic interactions of this type have earlier been described for both laboratory strains in mixed biofilm experiments [40], and biofilms formed by natural consortia [41,42,43]. The stimulation of sessile growth of biofilm deficient strains could be accomplished by both cell-to-cell interactions [44] and released products of the adherence-proficient strains [45,46]. According to Pathak [4] the microbial heterogeneity of the biofilm matrix promotes fulfilling of nutrient requirements and poses in the close vicinity the substrate and the microorganisms. The close contact between the plastic and the microorganisms by biofilm formation is known to favor the biodegradation process [47]. The here observed further delay in the rate of degradation could be explained by the hindering of metabolite fluid through the biofilm. The late stage increase in the degradation rate in the presence of the pure strains confirmed the observation that strain 7 and strain 2 do not form a good biofilm and increased degradation was connected with the increase in cell numbers in the culture. Such a suggestion was also confirmed by the higher protein concentration in free cells in these variants in comparison with MKC-P. At the same time the protein measured in biofilms was several times higher in MKC-P. SEM investigations confirmed surface damage in all variants, however biofilm formation only in MKC-P.

## 5. Conclusions

Our study provided evidence for an effective PCL degradation by a thermophilic community and *Brevibacillus thermoruber* strain 7 isolated from the same community. For the first time, a representative of the most important biotechnological group, namely bacilli, was isolated in the current work as a PCL degrader. The optimal temperature for the degradation process was the highest among the so far reported for PCL degradation that caused favorable changes in plastic properties. The demonstrated high degradation rate by strain 7 defined it as one of the best PCL degraders described, and community’s degradation rate was even higher. A possibility for full plastic degradation for four weeks by MKC-P and 63.6% by the pure strain and significant changes observed by SEM on the PCL surface suggested an effective application of the isolated community and the pure strain in plastic waste treatment.

## Figures and Tables

**Figure 1 biomolecules-11-01488-f001:**
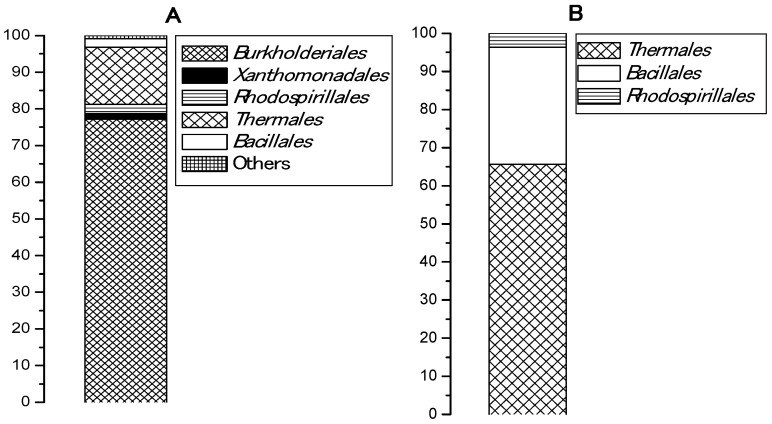
Bacterial orders represented (%) in: (**A**), MKC-C; (**B**), MKC-P.

**Figure 2 biomolecules-11-01488-f002:**
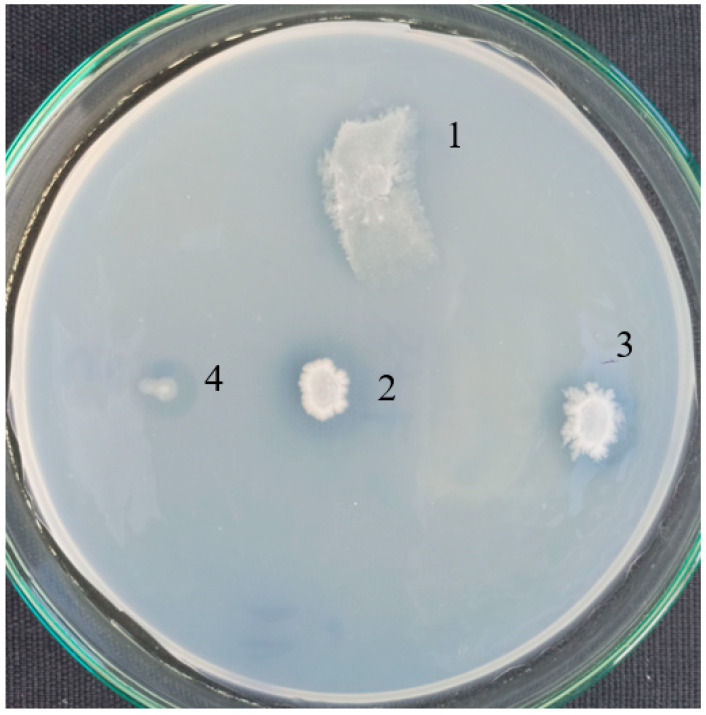
Clear halo by PCL-degrading bacteria isolated from Marikostinovo. 1, MKC-P; 2, *Brevibacillus thermoruber* strain 7; 3, *Aneurinibacillus thermoaerophilus* strain 2; 4, *Meiothermus cateniformans* strain 12.

**Figure 3 biomolecules-11-01488-f003:**
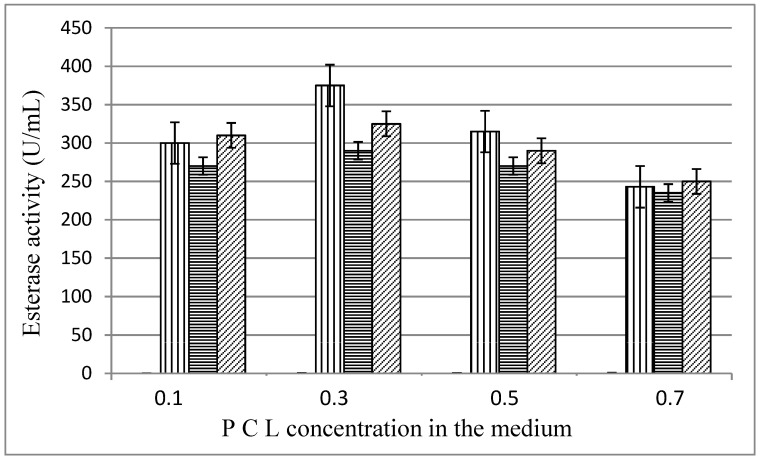
Esterase activity (48 h) in different PCL concentration in the medium; 
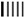
, MKC-P; 
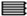
, *B. thermoruber* strain 7; 
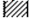
, co-culture of strain 7 and *A. thermoaerophilus* strain 2.

**Figure 4 biomolecules-11-01488-f004:**
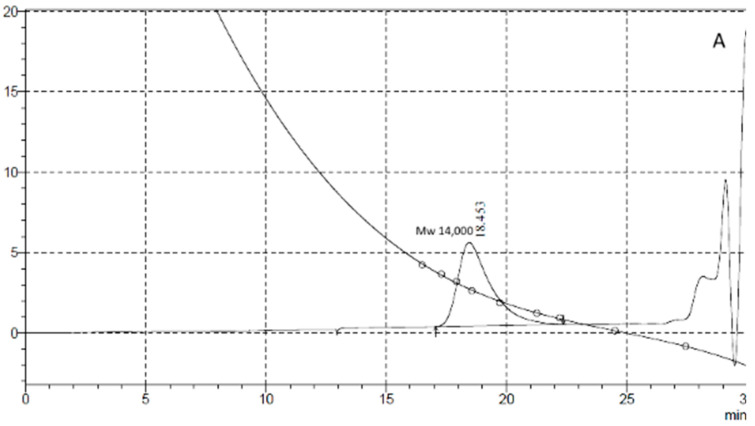
GPC elution pattern of PCL degradation intermediates on 48 h of cultivation. (**A**), Control—pure PCL; (**B**), 1, MKC-P; 2, co-culture of *B. thermoruber* strain 7 and *A. thermoaerophilus* strain 2; 3, *B. thermoruber* strain 7.

**Figure 5 biomolecules-11-01488-f005:**
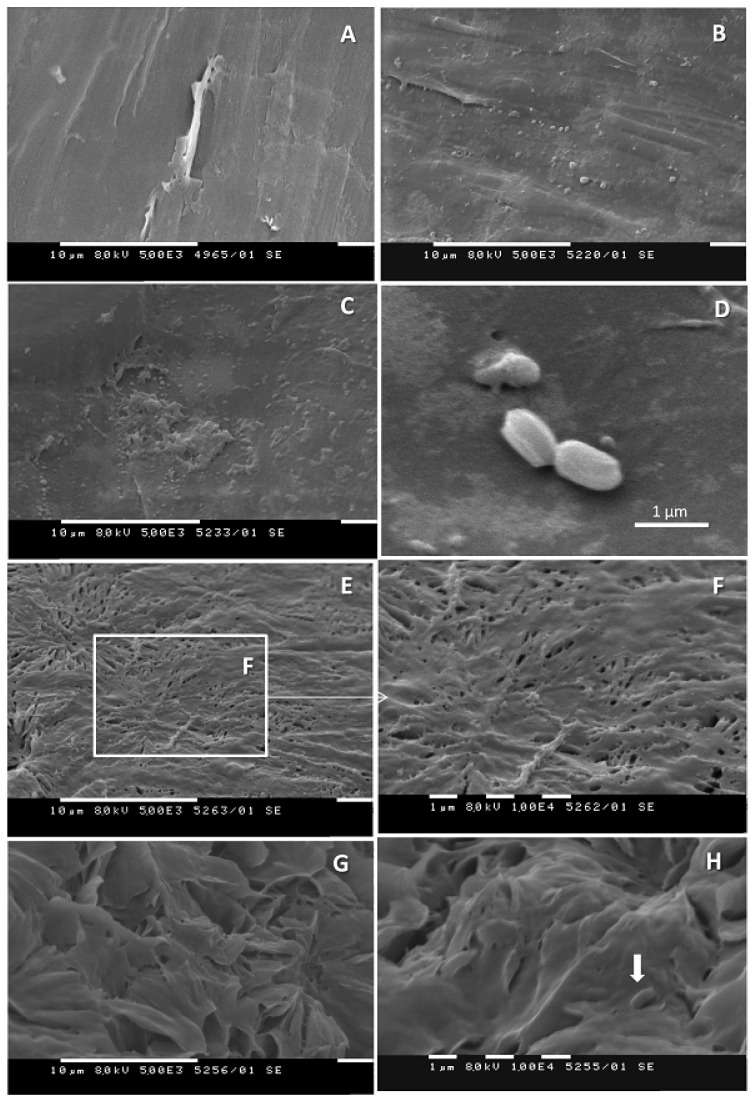
Changes of the surface of PCL pearls cultivated in the presence of *Brevibacillus thermoruber* strain 7. (**A**), control. (**B**–**D**), cultivation in the presence of the strain for 72 h. Deformations of the PCL surface comprise bubble-like elevations (**B**,**C**), single adherent bacteria are occasionally observed (**D**). (**E**–**H**), cultivation in the presence of the strain 7 for 3 weeks. The PCL surface is characterized by shallower (**E**,**F**) or deeper (**G**,**H**) infolds, with low incidence of attached bacteria (**H**, arrow).

**Figure 6 biomolecules-11-01488-f006:**
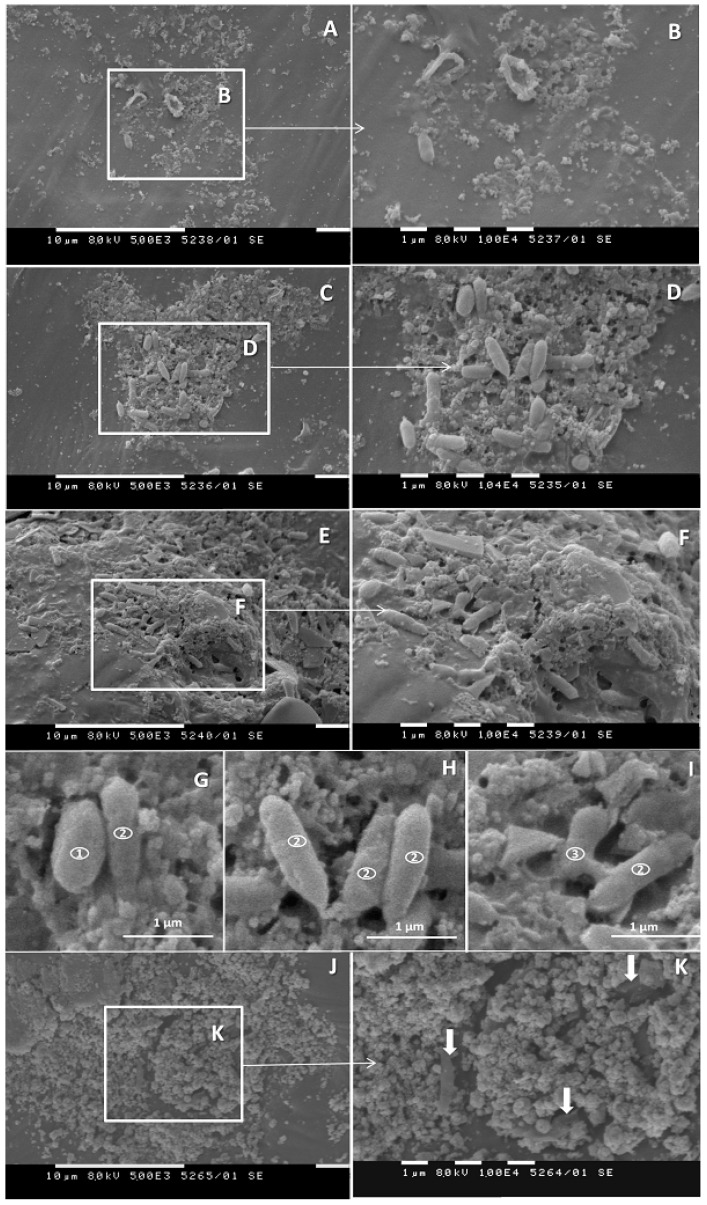
Biofilm formation on the surface of PCL pearls cultivated in the presence of the bacterial community. (**A**–**F**), 72-h of cultivation. Adherence of bacterial cells (**A**,**B**), formation of microcolonies (**C**,**D**) and an apparent penetration of bacteria into the plastic (**E**,**F**) is illustrated. (**G**–**I**) Biofilm bacteria with three distinct types of morpholiogy were present on the PCL pearls: (1) oval cells morphologically similar to those of B. thermoruber strain 7 (Figure 5D); (2) rod-like cells; (3) pyriform bacteria. (**J**,**K**). On week 3 areas with the presence of grain-like material predominated, and the bacterial cells were submerged under this material (**K**, arrows).

**Table 1 biomolecules-11-01488-t001:** Taxonomic groups identified in MKC-C.

Identified Group	Phylogenetic Level	Reads, %
*Caldimonas* sp.	Phylum *Proteobacteria*, Order *Burkholderiales*	42.2
*Tepidimonas ignava*	Phylum *Proteobacteria*, Order *Burkholderiales*	21.0
*Meiothermus* sp.	Phylum *Deinococcus-Thermus*, Order *Thermales*	14.5
*Tepidimonas* sp.	Phylum *Proteobacteria* Order *Burkholderiales*	13.7
*Bacillales*	Phylum *Firmicutes*, Order *Bacillales*	2.3
*Thermomonas* sp.	Phylum *Proteobacteria*, Order *Xanthomonadales*	1.5
*Elioraea tepidiphila*	Phylum *Proteobacteria* Order *Rhodospirillales*	1.1
*Thermaceae*	Phylum *Deinococcus-Thermus*, Order *Thermales*	1.1
*Roseomonas*	Class *Alphaproteobacteria* Order *Rhodospirillales* Family *Acetobacteraceae*	0.9
*Acetobacteraceae*	Class *Alphaproteobacteria* Order: *Rhodospirillales*	0.5
*Burkholderiales*	Phylum *Proteobacteria* Order: *Burkholderiales*	0.3

**Table 2 biomolecules-11-01488-t002:** Taxonomic groups identified in MKC-P.

Identified Group	Phylogenetic Affiliation	Reads (%)
*Meiothermus*	Phylum *Deinococcus-Thermus*, Order *Thermales*	58.1
*Brevibacillus*	Phylum *Firmicutes* Order *Bacillales* Family *Paenibacillaceae*	18.5
*Bacillales*	Phylum *Firmicutes*, Order *Bacillales*	11.6
*Thermaceae*	Phylum *Deinococcus-Thermus*, Order *Thermales*	7.5
*Elioraea tepidiphila*	Phylum *Proteobacteria* Order *Rhodospirillales*	3.7
*Paenibacillus*	Phylum *Firmicutes* Order *Bacillales*	0.4
*Bacillaceae*	Phylum *Firmicutes*	0.2

**Table 3 biomolecules-11-01488-t003:** Phylogenetic affiliation of the selected strains and esterase activity on 48 h of cultivation of the MKC-P and isolated strains alone and in a combination with *B. thermoruber* strain 7.

Highest-Homology Organism (Maximum % Identity of 16S rRNA Gene Sequence)	Sequence Number	Esterase Activity of Pure Strains (U/mL)	Esterase Activity in Co-Cultures (U/mL) with *B. thermoruber*
*Aneurinibacillus thermoaerophilus* strain 2	MW927323	115	325
*Brevibacillus thermoruber* strain 7	MW541896	290	-
*Meiothermus cateniformans* strain 12	MW927332	42	70
MKC-P	PRJNA766622		375

**Table 4 biomolecules-11-01488-t004:** Efficiency of PCL degradation by MKC-P, *B. thermoruber* strain 7, and *B. thermoruber* strain 7 + *A. thermoaerophilus* strain 2.

Microorganisms	Week	Final Weight	Weight Loss per 7 Days (mg)	PCL Degradation Rate (mg/d)	Weight Loss per 7 Days (%)	Protein Contents (mg/mL)			Esterase Assay U/mL
						Protein in Free Cells	Protein in Biofilms	Total Protein	
Control	4	150 ± 9	0	0	0	0	-	0	0
Community	1	88.2	61.8	8.83	41.2	0.23	1.31	1.54	375
	2	48.8	39.4	5.63	26.3	0.45	1.56	2.01	375
	3	20.1	28.7	4.1	19.1	0.63	1.39	2.02	333.8
	4	0	20.1	2.87	13.4	0.17	0.84	1.01	291.6
*B. thermoruber* strain 7	1	134.7	15.3	2.18	10.2	0.75	0.05	0.80	290
	2	115.2	19.5	2.78	13	0.78	0.15	0.93	290
	3	84.2	31.0	4.42	20.6	0.84	0.19	1.03	258
	4	54.5	29.7	4.24	19.8	0.11	0.14	0.25	125
*B. thermoruber* strain 7 + *A. thermoaerophilus* strain 2	1	125	25	3.57	16.7	0.82	0.13	0.95	325
	2	96.8	28.2	4.02	18.8	1.06	0.19	1.25	308
	3	67.6	29.2	4.17	19.5	1.18	0.19	1.37	280
	4	37.7	29.9	4.27	19.9	0.72	0.17	0.89	248

## Data Availability

Not applicable.

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
