# Peer review of "Degradation of Poly(ε-caprolactone) by a Thermophilic Community and Brevibacillus thermoruber Strain 7 Isolated from Bulgarian Hot Spring"

_biomolecules, 2021, doi:10.3390/biom11101488_

Round 1
Reviewer 1 Report
See attached file.

Author Response
Thank you for your valuable recommendations that we have taken in mind:
The authors attempted to isolate thermophilic PCL-degraders from environmental sources and characterize their PCL degrading capacity in this manuscript. The methods used for the enrichment and isolation of PCL degrading bacteria are sound. The formation of halo was used as indication of PCL degrader. Esterase activity was used to evaluate the effect of pH on biodegradation. The changes in protein content were used to monitor microbial growth. The authors used gravimetric weights, GPC analysis to characterize biodegradation.
The authors enriched PCL-degrading cultures at thermophilic condition (55 oC) and then isolated three bacterial strains with PCL degrading ability with Brevibacillus thermoruber strain 7 as the best. They described synergistic interactions by comparing MKC-P, B. thermoruber strain 7, and co-culture of . B. thermoruber strain 7 plus Aneurinibacillus thermoaerophilus strain 2. They found significant community difference between MKC-C and MKC-P cultures after enrichment with PCL (MKC-P), identified PCL-degrading strains with GenBank numbers.
This manuscript is well written and can be accepted for publication. Minor revision is needed as I described below.
Abstract:
- Indicate PCL is one of major biodegradable plastics with promising application to replace existing non-biodegradable polymers.
Such a sentence was added in the Abstract section.
- Indicate the temperature tested for strain 7.
The temperature was indicated.
Line 36. Cite updated data of global plastics production from PlasticsEurope
The site of PlasticsEurope was reached on 27.09.2021. The information in “Plastics - the Facts 2020” in the section “market-data” the last cited data are: World 2018 - 359 million tonnes; 2019 – 368 million tonnes (https://www.plasticseurope.org/application/files/3416/2270/7211/Plastics_the_facts-WEB-2020_versionJun21_final.pdf).
Line 60. Currently, PCL is still not a cost-effective polymer with $6-8/kg market price. Great efforts have focused on mass production at lower costs.
Such a sentence was added.
Line 61-69. The authors should cite more recent review and research papers on PCL biodegradation, e.g. Environ. Sci. Technol. 2018, 52, 10441-10452; Scientific Reports, 2019, 9:20236 https://doi.org/10.1038/s41598-019-56757-5.
Unfortunately, I do not know how to cite the noted paper. It concerns film composited by polycaprolactone (PCL) and grapefruit seed extract (GSE) as an antimicrobial agent against the pathogen Listeria monocytogenes. It does not concern information for PCL biodegradation.
Line 124. To isolate PCL-degrading bacteria, how long did you enrich PCL degrading microbes in MKC-P medium? Five days? Please indicate it.
The information on the cultivation is several rows below (l. 146-147) – “Cultures were incubated in the basal medium for 72 h on a shaker (80 rpm) at 55°C.” As the esterase activity did not changed between 48 and 72 h of cultivation we did not continue further.
Table 4. It is better to present the molecular weight of respective peaks. The MW distribution of original PCL should be presented as control.
Fig. 4 (I suggest the reviewer had in mind Fig. 4 instead Table 4). The MW distribution of original PCL was included in the figure as a subfigure “A”. The molecular weight of the respective peaks was included.
In discussion section. The authors may discuss further tests of biodegrading ability for PLA, LDPE, PS and PET using the isolates especially B. themoruber strain 7.
The same community MKC has shown an esterase activity against PS in our preliminary investigation however 6 fold lower than that against PCL. That’s why we chose PCL degradation as an object in the current manuscript. We do not know which strain carried PS activity, however testing B. themoruber strain 7 is a good idea for a future work.
Reviewer 2 Report
General Comments unnecessary
The paper on thermophilic degradation of poly(ϵ-caprolactone) presents a few interesting observations about the process, but doesn’t provide a great deal of insight into the process, and it definitely fails to describe the importance of the observations, or how the information disclosed in the paper could be applied to reduction of contamination by this commonly used polymer. The paper can be boiled down to a single statement: A thermophilic bacterial community was able to degrade 100% of added poly(ϵ-caprolactone), and the intact community was more capable of degradation than individual isolates from that community.
There is evidence presented that addition of the PCL to samples from the hot spring resulted in enrichment of the competent strains. I think the conclusions from the paper can be stated much more simply than the authors have done. Again, this is interesting observational work, but aside from the observation that thermophiles seem to degrade the plastic more effectively than might be expected from communities with a lower temperature optimum, there is little that is new.
I realize that English is not the first language of these authors. While I was able to follow the writing and understand the intent of the authors throughout the paper (with only a couple of minor exceptions), a good editing by journal staff or a copy editor as a consultant, would make the paper easier to follow for the reader – especially one who might not be familiar with degradation literature.
The sections on metagenomics and community structure do not contribute anything to the paper, except possibly that the addition of the plastic enriched a degrading community. There is almost no useful information in the metagenomics, and much of the section (methods, results, and discussion) is just a detailing of the taxa found without much of any relevance to the main topic of the paper. This type of analysis has become wildly popular, but rarely supports an understanding of anything about function of the communities. I suggest the community structure analyses be deleted. The paper will be much more succinct and the impact of the findings about the degradation will not be diluted by many many words that have little to do with the main topic.
Specific Comments:
- 61-62. I like that the authors included a statement that defines biodegradation. Too many readers, unless told otherwise, equate biodegradation with mineralization. They are not the same, and the statement by these authors is consistent with their findings reported later in the paper.
- 237-242. Given the isolation and culture conditions for the PCL degrading organisms, the effort to name them as part of the characterization is superfluous. The sequencing effort does indicate that the isolates were different one from another, and in reality that is all the information we need. I have no problem including the “names” of the isolates, but the authors need to recognize that they are not terribly informative except to discriminate among the isolates (the organisms could have been named with strain numbers to the same effect). I am not requesting changes in these lines. However, if the metagenomic “community structure” information is deleted as recommended above, the sentence in lines 246 and 247 must also be deleted. Indeed, the remainder of the paper would be much easier to read if, after naming the isolates, the authors referred to them by a simple strain number. I find plodding though multiple occurrences of “Aneurinibacillus thermoaerophilus strain 2 and Meiothermus cateniformans strain 12,” etc., etc. to be onerous; it could be simplified with no loss of information. (Should not genus and species names be italicized?)
- 97. I’m not exactly sure what is meant by “low-mineralization,” but I think the authors mean “low dissolved-mineral content.” Mineralization means something quite different, and the suggested correction would clarify.
- 105. Correct the beginning of the line to read “as a sole-carbon source.” Also, how was the sterilization process of the plastic beads confirmed?
- Section 2.2 can be deleted.
- 180. “6-digit accuracy” is vague. I don’t know which 6 digits we are talking about here. Replace with a phrase like “a balance with 10 μg accuracy,” or whatever the actual value was.
- 187. Please define DR as the degradation rate in this formula.
- 194. “The sample was resolved” Here I think: “ The dried sample was dissolved in…” or redissolved.
- 196. This one, I don’t understand. What is a “salinized” vial?
- 200. This is just a word-order problem, change to read “The GPC column set used was…”
- Table 1 and Figure 1 can be deleted with the material about the taxa in the communities.
- Table 3 needs serious editing and reformatting. I’m not sure what is controlling the margins here, but if they are in conformance to the Journal’s page design, perhaps rotating the table would make it easier to follow.
- The first paragraph can be deleted consistent with my comments in the General Comment section.
- 531. To what does “their” refer?
Author Response
General Comments unnecessary
The paper on thermophilic degradation of poly(ϵ-caprolactone) presents a few interesting observations about the process, but doesn’t provide a great deal of insight into the process, and it definitely fails to describe the importance of the observations, or how the information disclosed in the paper could be applied to reduction of contamination by this commonly used polymer. The paper can be boiled down to a single statement: A thermophilic bacterial community was able to degrade 100% of added poly(ϵ-caprolactone), and the intact community was more capable of degradation than individual isolates from that community.
There is evidence presented that addition of the PCL to samples from the hot spring resulted in enrichment of the competent strains. I think the conclusions from the paper can be stated much more simply than the authors have done. Again, this is interesting observational work, but aside from the observation that thermophiles seem to degrade the plastic more effectively than might be expected from communities with a lower temperature optimum, there is little that is new.
I realize that English is not the first language of these authors. While I was able to follow the writing and understand the intent of the authors throughout the paper (with only a couple of minor exceptions), a good editing by journal staff or a copy editor as a consultant, would make the paper easier to follow for the reader – especially one who might not be familiar with degradation literature.
The sections on metagenomics and community structure do not contribute anything to the paper, except possibly that the addition of the plastic enriched a degrading community. There is almost no useful information in the metagenomics, and much of the section (methods, results, and discussion) is just a detailing of the taxa found without much of any relevance to the main topic of the paper. This type of analysis has become wildly popular, but rarely supports an understanding of anything about the function of the communities. I suggest the community structure analyses be deleted. The paper will be much more succinct and the impact of the findings of the degradation will not be diluted by many many words that have little to do with the main topic.
According to the authors’ opinion, an important element of plastic degradation research is the revealing of microorganisms included in the process. Knowledge on the flourishing community taxa in the presence of plastics envisages the nature of the isolates that should be searched for further isolation. The ratio between different community members could suggest their role in biofilm formation as some of the microorganisms could not have enough capacity to degrade the plastic but could contribute biofilm formation posing the substrate and enzyme producer in close vicinity (Meiothermus in our case). The results from the metagenome analysis also contribute to metagenome screening for desired esterase activities from the preferred microbial group. Functional metagenomics is an object of our future work together with a description of enzymes and genes from active community members. Additionally, such information contributes to the design of the community for successful plastic waste treatment. Plastic active taxa in psychrophilic communities are already known as a result of several works on the comparison between metagenomes from polluted and unpolluted seawater. Still, the information for thermophilic communities is missing, and the way to reach the microorganisms that could contribute to this process is just a metagenomic analysis. We fully agree with the reviewer statement “A thermophilic bacterial community was able to degrade 100% of added poly(ϵ-caprolactone), and the intact community was more capable of degradation than individual isolates from that community”, however, it is important to know the structure of this thermophilic community that provides 100% degradation in order to apply it for reduction of contamination by this commonly used polymer.
English was checked again by an international translator.
Specific Comments:
- 61-62. I like that the authors included a statement that defines biodegradation. Too many readers, unless told otherwise, equate biodegradation with mineralization. They are not the same, and the statement by these authors is consistent with their findings reported later in the paper.
Thank you for this assessment.
- 237-242. Given the isolation and culture conditions for the PCL degrading organisms, the effort to name them as part of the characterization is superfluous. The sequencing effort does indicate that the isolates were different one from another, and in reality that is all the information we need. I have no problem including the “names” of the isolates, but the authors need to recognize that they are not terribly informative except to discriminate among the isolates (the organisms could have been named with strain numbers to the same effect). I am not requesting changes in these lines. However, if the metagenomic “community structure” information is deleted as recommended above, the sentence in lines 246 and 247 must also be deleted. Indeed, the remainder of the paper would be much easier to read if, after naming the isolates, the authors referred to them by a simple strain number. I find plodding though multiple occurrences of “Aneurinibacillus thermoaerophilus strain 2 and Meiothermus cateniformans strain 12,” etc., etc. to be onerous; it could be simplified with no loss of information. (Should not genus and species names be italicized?)
In the current version strain numbers were used in most cases an exception of their first mentioning in each section, tables, and figures according to the requirements in microbiological papers. Sorry for the not italicized names, an error occurred during the formatting of the file. This time will submit it as pdf file. Hope this will not happen during the submission of the current version.
- 97. I’m not exactly sure what is meant by “low-mineralization,” but I think the authors mean “low dissolved-mineral content.” Mineralization means something quite different, and the suggested correction would clarify.
“low mineralization” was replaced by “low dissolved-mineral content.”
- 105. Correct the beginning of the line to read “as a sole-carbon source.” Also, how was the sterilization process of the plastic beads confirmed?
The sentence was corrected according to the reviewer's suggestion. Controls containing plastic pearls without inoculum were included in each experiment in the aim to confirm not only the sterilization process but also to follow a possible influence of temperature on the plastic changes.
- Section 2.2 can be deleted.
The importance of these investigations according to the authors has been discussed above.
- 180. “6-digit accuracy” is vague. I don’t know which 6 digits we are talking about here. Replace with a phrase like “a balance with 10 μg accuracy,” or whatever the actual value was.
The phrase was replaced according to the reviewer’s recommendation.
- 187. Please define DR as the degradation rate in this formula.
DR was defined as a degradation rate in the formula
- 194. “The sample was resolved” Here I think: “ The dried sample was dissolved in…” or redissolved.
The term “resolved” was replaced by “dissolved”.
- 196. This one, I don’t understand. What is a “salinized” vial?
“salinized” was removed.
- 200. This is just a word-order problem, change to read “The GPC column set used was…”
The sentence was changed this way: The GPC Column Set PSS SDV 50Å (300mm x 8.00 mm x 5 μm), PSS SDV 100Å (300 mm x 8.00 mm x 5 μm), PSS SDV Linear М (300 mm x 8.00 mm x 5 μm) were used.”
- Table 1 and Figure 1 can be deleted with the material about the taxa in the communities.
Table 1 and Figure 1 are connected with the metagenome analysis. The authors’ opinion for its importance was discussed above.
- Table 3 needs serious editing and reformatting. I’m not sure what is controlling the margins here, but if they are in conformance to the Journal’s page design, perhaps rotating the table would make it easier to follow.
Table 3 was rotated
- The first paragraph can be deleted consistent with my comments in the General Comment section.
The authors’ opinion on the importance of the metagenomic analysis is discussed above.
- 531. To what does “their” refer?
“their” was replaced by “an effective application of the isolated community and the pure strain”.
Reviewer 3 Report
In this work, the authors describe the Poly(ε-caprolactone) degradation ability of thermophilic communities. I think that is very nice work, here my consideration:
- I suggest that the authors pay attention to the affiliations; the numbers must go to the top
- Line 44 (C6H10O2)n , and also look at in the full test
- I suggest writing the names of the microorganisms in italics
Author Response
Thank you for the careful reading. All noted errors were removed.
In this work, the authors describe the Poly(ε-caprolactone) degradation ability of thermophilic communities. I think that is very nice work, here my consideration:
- I suggest that the authors pay attention to the affiliations; the numbers must go to the top
Affiliation numbers were corrected
- Line 44 (C6H10O2)n , and also look at in the full test
The numbers were placed as subscript
- I suggest writing the names of the microorganisms in italics
Sorry for the not italicized names, an error occurred during the formatting of the file. This time will submit it as pdf file.
Reviewer 4 Report
The authors in this manuscript provided a proof-of-concept for the degradation of Poly(ε-caprolactone). This is a relevant study and is totally aligned with UN goals.
I only have minor comments relatively to the quality of presentation. For instance most of the graphics appears to be done using excel. The authors should try to use other softwares and improved in this way their figures.
Also some figure legends appears to be incomplete e.g. 3 and 4.
It would also be very helpful if the authors include in this manuscript a schematic figure.
Author Response
Thank you for the valuable suggestions. Here are the reviewers' answers:
The authors in this manuscript provided a proof-of-concept for the degradation of Poly(ε-caprolactone). This is a relevant study and is totally aligned with UN goals.
I only have minor comments relative to the quality of the presentation. For instance most of the graphics appears to be done using excel. The authors should try to use other softwares and improved in this way their figures.
Now figs. 1 and 3 were prepared by Origin software, figs. 2, 5, and 6 are pictures, fig. 4 was drawn from the column software.
Also, some figure legends appear to be incomplete e.g. 3 and 4.
Now figure 3 and 4 legends are complete.
It would also be very helpful if the authors include in this manuscript a schematic figure.
We envisage preparing a schematic figure on the full biodegradation process in our next work when the enzymes participating will be included.
Round 2
Reviewer 2 Report
Most of the genus and species names are italicized in this iteration of the manuscript, but they are not in Table 3. Give the paper a good going over to be sure these names are italicized consistently throughout.
Author Response
Thanks for all requirements that contribute a good going over of the work. Sorry for the omission of italicized species names in the title of Table 3. The manuscript was checked again carefully and three other omissions were also corrected. They are noted by the "track changes" option.